# A Load Balancing Algorithm for Mobile Devices in Edge Cloud Computing Environments

**JongBeom Lim** [1] and **DaeWon Lee** [2,*]

[1] Department of Game & Multimedia Engineering, Korea Polytechnic University, Siheung-si, Gyeonggi-do 15073, Korea; jblim@kpu.ac.kr
[2] Department of Computer Engineering, Seokyeong University, Seongbuk-gu, Seoul 02713, Korea
* Correspondence: daelee@skuniv.ac.kr

**Abstract:** As current data centers and servers are growing in size by orders of magnitude when needed, load balancing is a great concern in scalable computing systems, including mobile edge cloud computing environments. In mobile edge cloud computing systems, a mobile user can offload its tasks to nearby edge servers to support real-time applications. However, when users are located in a hot spot, several edge servers can be overloaded due to suddenly offloaded tasks from mobile users. In this paper, we present a load balancing algorithm for mobile devices in edge cloud computing environments. The proposed load balancing technique features an efficient complexity by a graph coloring-based implementation based on a genetic algorithm. The aim of the proposed load balancing algorithm is to distribute offloaded tasks to nearby edge servers in an efficient way. Performance results show that the proposed load balancing algorithm outperforms previous techniques and increases the average CPU usage of virtual machines, which indicates a high utilization of edge servers.

**Keywords:** load balancing; edge cloud computing; graph coloring; genetic algorithm; cloud consolidation

## 1. Introduction

Load balancing is a fundamental problem in distributed systems and cloud computing environments [1–3]. The aim of load balancing is to distribute workloads to a set of reliable servers in the system. When workloads are distributed without load balancing techniques, the user may exhibit uncomfortable computing experiences, such as quality of service (QoS) and service level agreement (SLA) violations [4,5], unreliable data processing [6,7], and a lack of responsiveness [8,9]. As the scale of the state of the art computing systems grows in size, implementing load balancing techniques becomes more important in cloud computing environments [10,11].

To support real-time services and quick responsiveness for edge cloud computing users, the edge cloud computing paradigm has been developed to provide partial cloud computing services at the edge for a certain amount of computing resources [12–14]. Since the edge devices have limited capabilities and resources compared to the central cloud computing server, implementing load balancing techniques requires carefully tailored controls in edge cloud computing environments [15–17].

In this paper, we develop a load balancing technique in edge cloud computing environments with mobile devices. The proposed load balancing technique is based on a traditional graph problem called graph coloring [18–20] that minimizes the number of colors on vertices while satisfying a property (no adjacent vertices exhibit the same color). By analogy, the vertices of a graph can be considered as mobile devices, the edges of a graph can be considered as nearby discoverable mobile devices, and the distinctive graph colors are considered as distinctive edge servers.

To reduce the complexity of a load balancing algorithm, we employ a genetic algorithm approach [21–23] that is inspired by natural evolution and reflects the natural selection process, with a fitness function for individuals for producing the next generation. With this approach to implementing the proposed load balancing technique, we can achieve both low algorithm complexity and scalability.

The rest of the paper can be summarized as follows. Section 2 discusses our research background and describes the problem definition. Section 3 proposes our load balancing technique with graph coloring based on a genetic algorithm in edge cloud computing environments. Section 4 provides comparative performance evaluation results to show the effectiveness of load balancing. Finally, Section 5 concludes the paper.

## 2. Background and Problem Definition

In this section, we review the research background and describe the problem definition. As shown in Figure 1, there are three actors in edge cloud computing environments: (1) mobile devices, (2) edge servers, and (3) the central cloud server. Since the primary use of edge cloud computing environments is task offloading, the offloading framework resides in the three actors. The mobile devices can move around as needed and the tasks for the mobile devices get offloaded to a nearby edge server or the central cloud server.

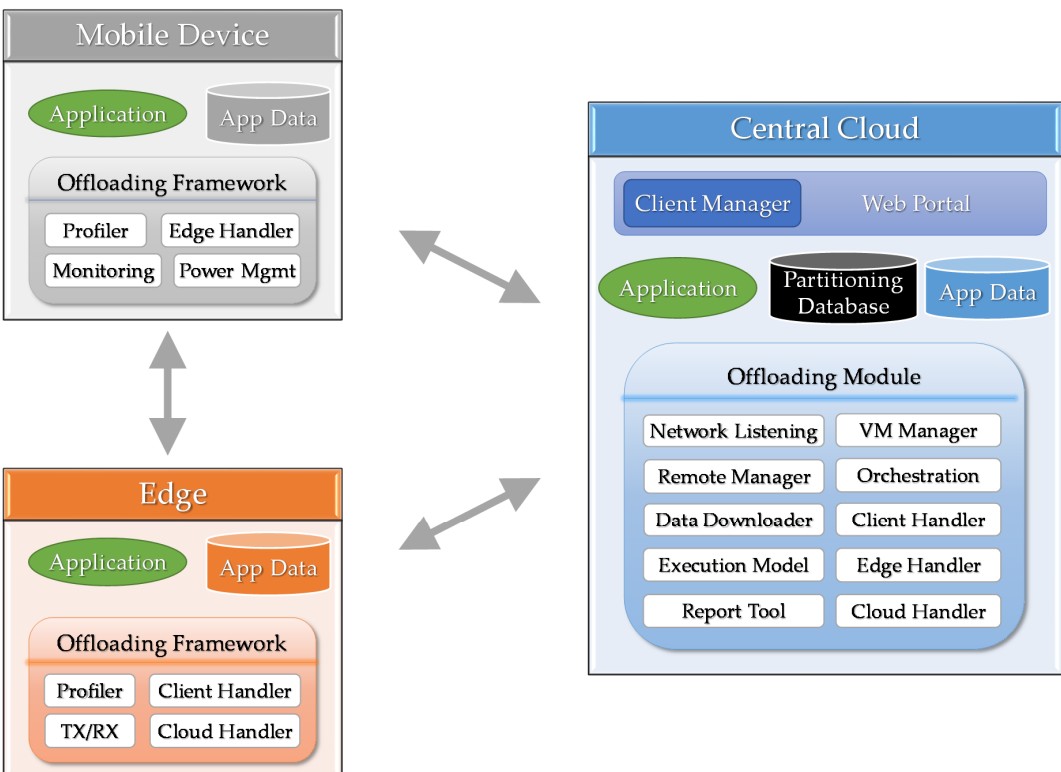

**Figure 1.** An edge cloud architecture and its interaction.

The main benefit of offloading to edge servers is that it can support real-time tasks by providing low latency connectivity for the application on mobile devices [24,25]. The problem arises when there are a number of mobile devices in hot spots. In such a scenario, multiple devices try to access nearby edge servers, which have limited capabilities compared to the central cloud server. When a load balancing technique is not applied, some edge servers are prone to failures.

However, implementing a load balancing technique for edge servers in edge cloud computing environments is not an easy task, and a few load balancing techniques are implemented for edge servers; previous load balancing techniques are mainly for virtual machine or container consolidation

and orchestration in the central cloud server [26,27]. Thus, we design and implement a load balancing technique in edge cloud computing environments with graph coloring based on a genetic algorithm.

## 2.1. Graph Coloring

Graph coloring in graph theory is a problem of assigning colors to the vertices and edges of a graph [18–20]. In this paper, we consider the graph coloring problem of assigning colors to vertices, such that no adjacent vertices of a graph exist of the same color. The object of graph coloring is to minimize the number of colors when assigning colors to vertices.

Figure 2 shows Petersen graphs [28,29] without and with coloring. In Figure 2a, all the vertices of the graph are the same, while there are three colors (orange, blue, and green) in Figure 2b. Notice that no adjacent vertices of the graph are the same. Although we can assign more colors to the graph to avoid violating the property (i.e., to make sure that no adjacent vertices are of the same color), we assign three since the objective is to minimize the number of colors. If we try to use two colors instead of three, the property cannot be satisfied (i.e., there exist adjacent vertices with the same color).

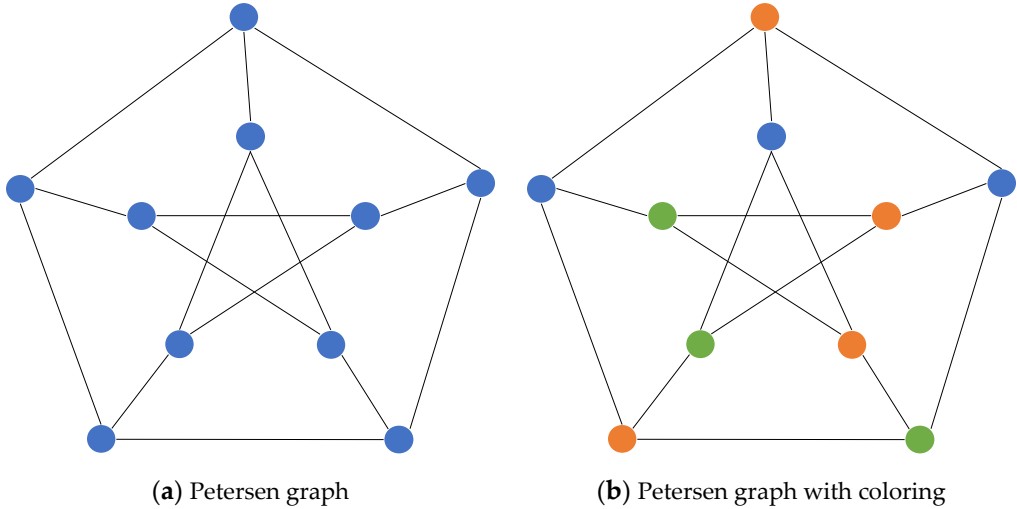

(**a**) Petersen graph        (**b**) Petersen graph with coloring

**Figure 2.** Petersen graphs without and with coloring.

We apply graph coloring to our load balancing technique. By analogy, the vertices of a graph can be considered mobile devices, and the edges of a graph can be considered nearby discoverable mobile devices. The distinctive graph colors are considered as distinctive edge servers. The motivation of this approach is to distribute the tasks of mobile devices to multiple edge servers while minimizing the number of required edge servers. If we assign tasks from mobile devices that are closely located to unnecessary edge servers geometrically far apart from the mobile devices, the real-time requirement cannot be satisfied.

## 2.2. Genetic Algorithm

A genetic algorithm [21–23] is a nature-inspired search heuristic inspired by Charles Darwin's theory. To mimic natural evolution processes, it reflects the natural selection process with a fitness function for individuals and produces the next generation based on crossover and mutation processes.

Figure 3 shows the process of a genetic algorithm. It starts with the initial population of the genesis generation and uses a fitness function to measure scores for individuals in the population. Then, it selects a target based on the fitness function and processes the crossover and mutation processes to mimic the evolutionary process of humans.

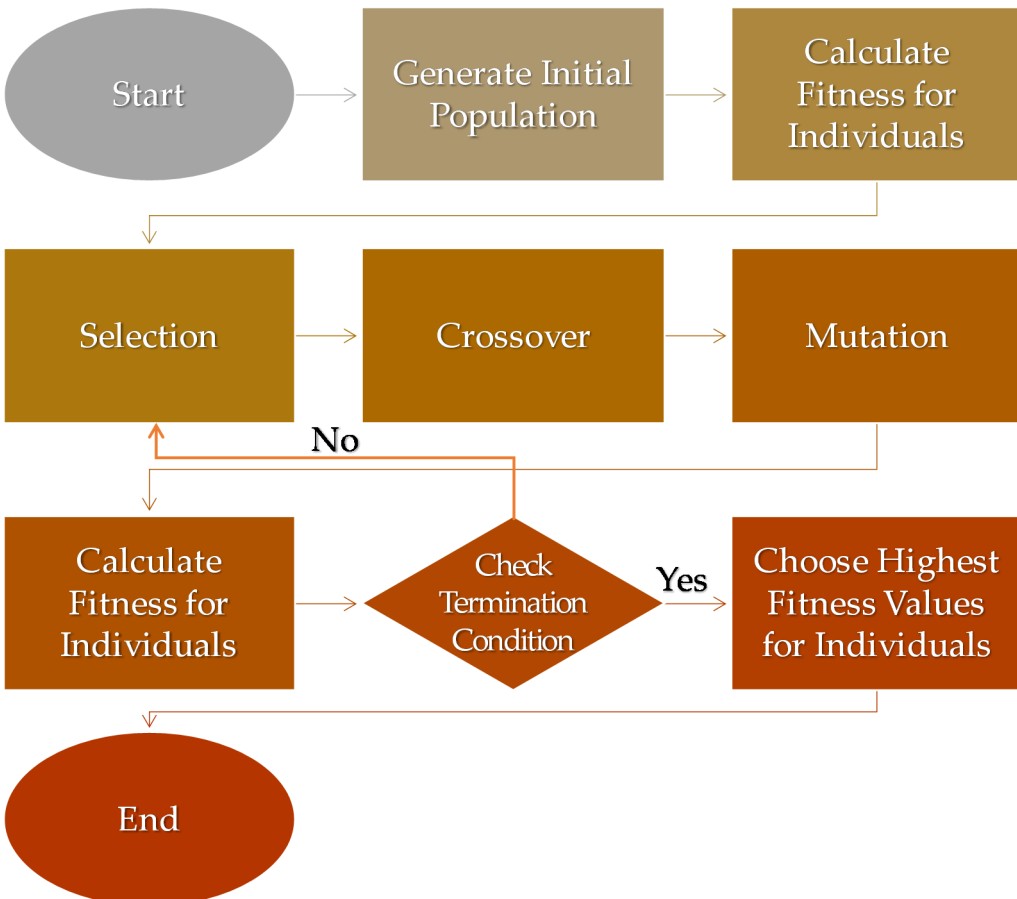

**Figure 3.** The process of a genetic algorithm.

Specifically, a genetic algorithm can be used to solve optimization problems for large datasets and search space. Unlike other strategies, a genetic algorithm uses a random approach starting from random initialization and it tries to find a solution by searching only in the space based on fitness scores. The fitness function is used for scoring each chromosome (solution candidate), and individuals that have the best fitness scores will be used for reproduction.

To survive from fitness, individuals try to find the region in which there are chances of getting a better fitness score. Then, they generate offspring (reproduction) by performing crossover and mutation processes. In the crossover process, parents generate offspring by selecting random points from the parents, while in the mutation process, some of the genes may have different values from the parents by mimicking the natural mutation. The reproduction process is iterated until the termination condition is met.

In order to implement the graph coloring problem, we use a genetic algorithm to reduce the complexity of the algorithm. In scalable systems like edge cloud computing environments, reducing the complexity of an algorithm is a crucial factor in terms of both deployment and implementation. The advantages of employing a genetic algorithm are as follows.

Firstly, it can solve fundamental problems with global maxima and minima. Secondly, it can handle complex mathematical representations with low complexity, while the linear programming approach exhibits high complexity. Thirdly, it supports problems that are difficult to represent in mathematical forms. Fourthly, it is resilient to data noise and failures. Fifthly, it is easy to implement solutions by exploiting parallel and distributed processing. Lastly, it can learn with new data and information in a continuous fashion to achieve global maxima and minima.

## 2.3. Literature Review

In [30], the authors proposed a hierarchical edge cloud architecture that distributes peak loads to different tiers of cloud servers. To adaptively distribute mobile workloads to different tiers, they developed a workload placement algorithm that monitors edge servers' computational capacity. In [31], the authors proposed a cooperative approach for two data centers. When a data center cannot accept a user's request, it forwards the request to another data center. The authors designed a mathematical model for the cooperative approach with a Poisson flow and recommended the dynamic tuning of system parameters.

In [32], the graph repartitioning theory is used to implement a load balancing algorithm that supports dynamicity. The authors constructed the system model of fog computing and developed the load balancing technique by considering vertices as representational nodes and edges as the data dependencies between tasks. Although the concept is similar to our approach using the graph theories, the implementation and the aim of the load balancing differ. While our method is to distribute offloaded tasks to nearby edge servers in an efficient way, the authors' approach is to minimize the migration overhead.

The authors of [33] developed an authentication technique in edge cloud computing environments. In terms of task allocation, their technique tries to find less overloaded edge servers by monitoring edge servers in the system and performing the breadth first search method to implement the load balancing technique. While the authors showed the effectiveness of the load balancing technique, the scalability and complexity issues of the algorithm remain unresolved.

Table 1 shows the feature comparison with the aforementioned studies. Our technique considers both workloads and edge servers for load balancing with mobility support. More importantly, the proposed technique scales well in terms of the size of mobile devices and the algorithm's complexity. When nearby edge servers are busy, our method forwards the request to the central cloud server. To minimize response time and support real-time applications, we consider nearby edge servers from the users and implement the algorithm based on the location of the mobile users.

**Table 1.** Feature comparison with previous studies.

| Feature | [30] | [31] | [32] | [33] | Proposed |
|---|---|---|---|---|---|
| Workload load balancing | ○ | ○ | ○ | ○ | ○ |
| Edge server load balancing | × | ○ | × | × | ○ |
| Support mobility | × | × | ○ | × | ○ |
| Scalability | △ | × | × | × | ○ |
| Support both edge servers and the central cloud | ○ | × | × | × | ○ |
| Support Real-time | × | ○ | × | × | ○ |
| Location-aware | × | × | ○ | × | ○ |

## 2.4. Problem Definition

When there are numerous mobile devices in hot spots and multiple devices trying to access nearby edge servers, which have limited capabilities, some edge servers are too busy to process the offloaded tasks of mobile devices, while some other edge servers can be idle. To mitigate this unbalanced problem at the edges, we design and implement a load balancing technique in edge cloud computing environments.

We define a load balancing problem as graph coloring. To distribute offloaded tasks to nearby edge servers, we consider the vertices of a graph as mobile devices, the edges of a graph as nearby discoverable mobile devices, and the graph colors as edge servers. In the graph coloring problem, the objective is to minimize the number of edge servers for a hot spot that prevents service level

agreement and quality of service violations. In the next section, we detail our load balancing technique with graph coloring based on a genetic algorithm in edge cloud computing environments.

## 3. The Proposed Load Balancing Algorithm

In this section, we describe the proposed load balancing technique with graph coloring based on a genetic algorithm. The design of the proposed technique considers the compatibility of edge cloud frameworks. Therefore, it can be integrated into existing edge cloud frameworks as a module.

Algorithm 1 details the proposed load balancing algorithm for edge servers in edge cloud computing environments. The input is $G = (V, E)$, where $V$ can be viewed as edge servers and $E$ can be viewed as nearby discoverable edge servers in a hot spot. Therefore, there is $G = (V, E)$ per hot spot in edge cloud computing environments. The output is a colored graph, that is, $G_{color} = (V_{color}, E)$.

There are several initialization variables. The *num_color* variable is for the number of colors of a graph. Since the objective is to minimize the number of colors for graph coloring, we choose a random number at the beginning. With the randomly generated value of the *num_color* variable, it performs the graph coloring algorithm. When it finds a solution without violating the property (i.e., no adjacent vertices are the same color), it tries with the decremented *num_color* (i.e., *num_color* = *num_color* − 1). If it cannot find a solution for graph coloring, it increases the *num_color* variable (*num_color* = *num_color* + 1). Then, it performs the algorithm again.

Because the proposed graph coloring algorithm is based on a genetic algorithm, we define the fitness function, and the *condition* and *max_generation* variables are for the termination condition of a genetic algorithm. After variable initialization, it calls the startJourney function (line 1). The startJourney function starts by initializing the *iteration* variable to 0, then generates the initial population by assigning random values to the vertices of a graph (lines 5–7).

After the initial population, it performs the graphColoring function as described in lines 26–33. The algorithm iteratively calls the graphColoring function by checking the *condition* and *iteration* variables (line 8). If the *condition* variable is *false* or the *iteration* variable does not reach *max_generation*, it calls the graphColoring function with the *G* and *num_color* parameters in the system. After returning from the graphColoring function, it assigns a new value of the *condition* variable (*true* if no adjacent vertices are the same color) and increases the *iteration* variable by 1 (lines 11–12).

If the *condition* variable is *true*, then it chooses the highest fitness values for individuals (i.e., the best color for each vertex of the graph) and decreases the *num_color* variable by 1 (lines 17–20). Then, it calls the startJourney function again, since there is a potential possibility that satisfies the graph coloring property with the decreased *num_color*. If the condition variable is *false*, it increases the *num_color* variable by 1 and calls another instance of the startJourney function.

When deploying the proposed load balancing algorithm, edge servers additionally check whether they can execute the offloaded tasks or not. If the edge server can support the offloaded tasks, it schedules for task execution. Otherwise, it forwards the offloaded tasks to the central cloud server with additional network traffic. This method prevents unnecessary task failures.

When deciding whether to forward tasks from edge servers to the central cloud server, the proposed technique checks the waiting time for edge servers and forwarding time for offloading tasks from edge servers to the central cloud server. When the waiting time for edge servers is less than the forwarding time for offloading tasks from edge servers to the central cloud server, the proposed technique does not forward the offloaded tasks to the central cloud. Otherwise, it forwards the offloaded tasks to the central cloud server.

## 4. Performance Evaluation

In this section, we present the results of the performance evaluation that demonstrate the efficiency of the proposed load balancing technique in edge cloud computing environments. In order to validate the load balancing performance, we implement the other three methods, that is, Cloud First, Round Robin, and Priority.

---

**Algorithm 1. The proposed load balancing algorithm based on graph coloring**

---

**Input:** $G = (V, E)$

**Output:** $G_{color} = (V_{color}, E)$

**Initialization:** *num_color* ← *random_number*;

*fitness_function* ← generateFitnessFunction();

*condition* ← *false*;

*max_generation* ← retrieveParameters(*generation*);

1:    **call** startJourney (*num_color*);

2:    **function** startJourney (*num_color*)

3:    *iteration* = 0;

4:    // Generate Initial Population

5:    **for all** $V_i \in G = (V, E)$ **do**

6:    $V_i$.*color* ← randomColor (*num_color*);

7:    **end for**

8:    **while** (*condition == false && iteration <= max_generation*)

9:    **call** graphColoring (*G, num_color*);

10:    // Check Termination Condition

11:    *condition* ← checkTerminationCondition();

12:    *iteration* ← *iteration* + 1;

13:    **end while**

14:    **if** *condition == true* **then**

15:    // Choose Highest Fitness Values for individuals

16:    **for all** $V_i \in G = (V, E)$ **do**

17:    $V_i$.*color* ← getColor (*fitness_function, num_color*);

18:    **end for**

19:    *num_color* ← *num_color* - 1;

20:    startJourney (*num_color*);

21:    **else**

22:    *num_color* ← *num_color* + 1;

23:    startJourney (*num_color*);

24:    **end if**

25:    **end function**

26:    **function** graphColoring (*G, num_color*)

27:    // Calculate Fitness for Individuals

28:    evaluateFitness (*G = (V, E), fitness_function*);

29:    // Selection, Crossover, and Mutation

30:    selection (*G = (V, E), fitness_function*);

31:    crossover (*G = (V, E), fitness_function*);

32:    mutation (*G = (V, E), fitness_function*);

33:    **end function**

---

In the Cloud First method, mobile devices prefer the central cloud server over edge servers nearby. In other words, they contact the central cloud server whenever it is available and access edge servers if the central cloud server is busy. In the Round Robin method, mobile devices select edge servers in a round-robin fashion for offloading tasks.

The Priority method calculates weight values for the edge servers and the central cloud server and offloads tasks to whichever has the best cost value. For example, once a mobile device decides to offload a task, it prioritizes the weight values for edge servers (lower weight value) and the central cloud server (higher weight value) and calculates the total cost by considering the task length and the type.

Our proposed method adds a load balancing technique to select edge servers for offloading tasks of mobile devices in edge cloud environments. We measure the task waiting time, network traffic, the average CPU usage of virtual machines, and the number of tasks that are successfully

executed on edge servers. In order to show the scalability of the techniques, we increase the number of mobile devices from 200 to 1000 in increments of 200. A mobile device generates five to nine tasks for offloading and we assume there are sufficient edge servers in edge cloud environments.

For types of tasks, we consider three applications: augmented reality, health monitoring, and computation-intensive data processing. Since the types of the three applications are different, the properties of the three applications are also dissimilar. For example, the augmented reality application requires low latency, and the health monitoring application sends sensor data and the server examines (processes) the data and forwards the results to the application in a timely fashion. The computation-intensive data processing application sends programming codes to the server, and the server returns the results afterward. Regarding this, we configure a mobile device to generate random tasks, five to nine at a time, according to the types of applications [34]. All of the tasks from mobile devices in a hotspot are distributed according to the policies of the algorithms; in our case, the tasks are evenly distributed to nearby edge servers up to the number of colors.

Figures 4–8 show the performance comparison results when the numbers of mobile devices are 200, 400, 600, 800, and 1000. When the number of mobile devices is 200, the task waiting time (Figure 4a) of the proposed load balancing technique is the shortest (24 ms), and those of other three techniques are 208 ms, 46 ms, and 54 ms for Cloud First, Round Robin, and Priority, respectively. For the Cloud First method, the waiting time is relatively long since many tasks try to access the central cloud server at the same time.

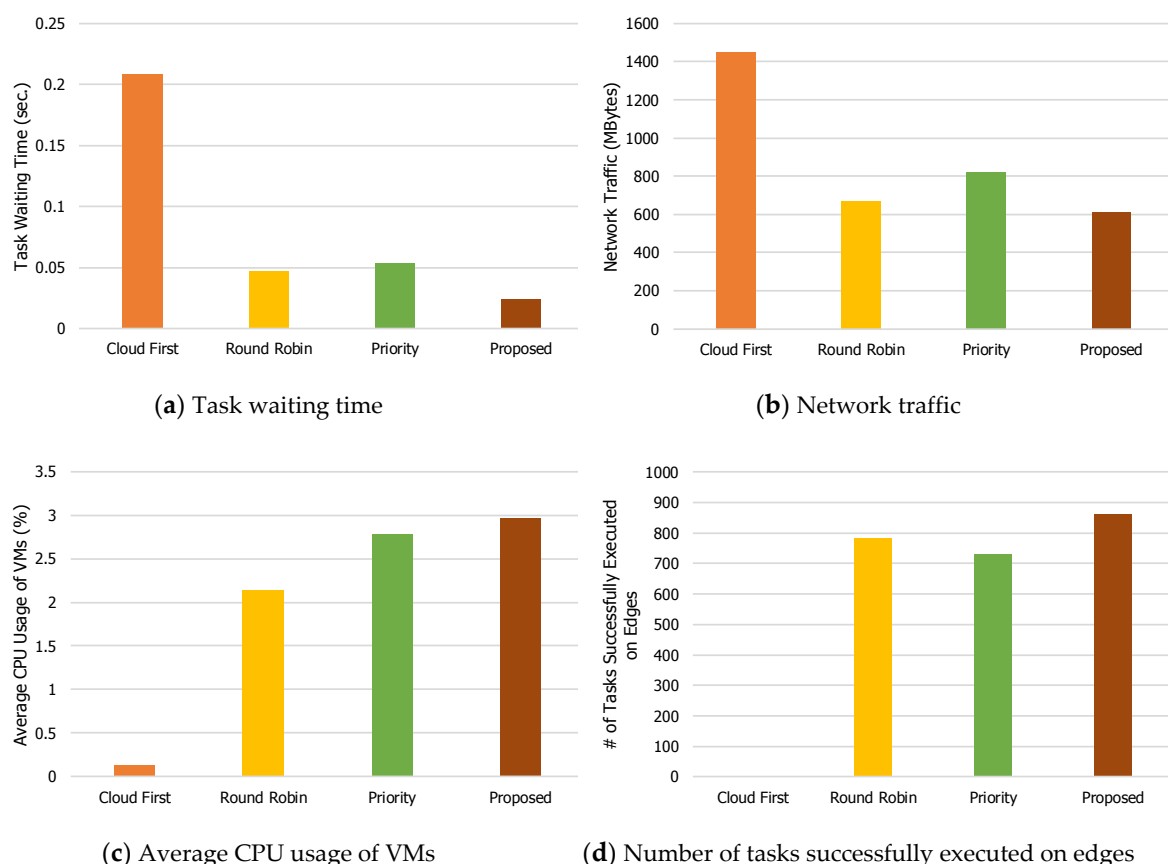

(**a**) Task waiting time

(**b**) Network traffic

(**c**) Average CPU usage of VMs

(**d**) Number of tasks successfully executed on edges

**Figure 4.** Performance comparisons when the number of mobile devices is 200.

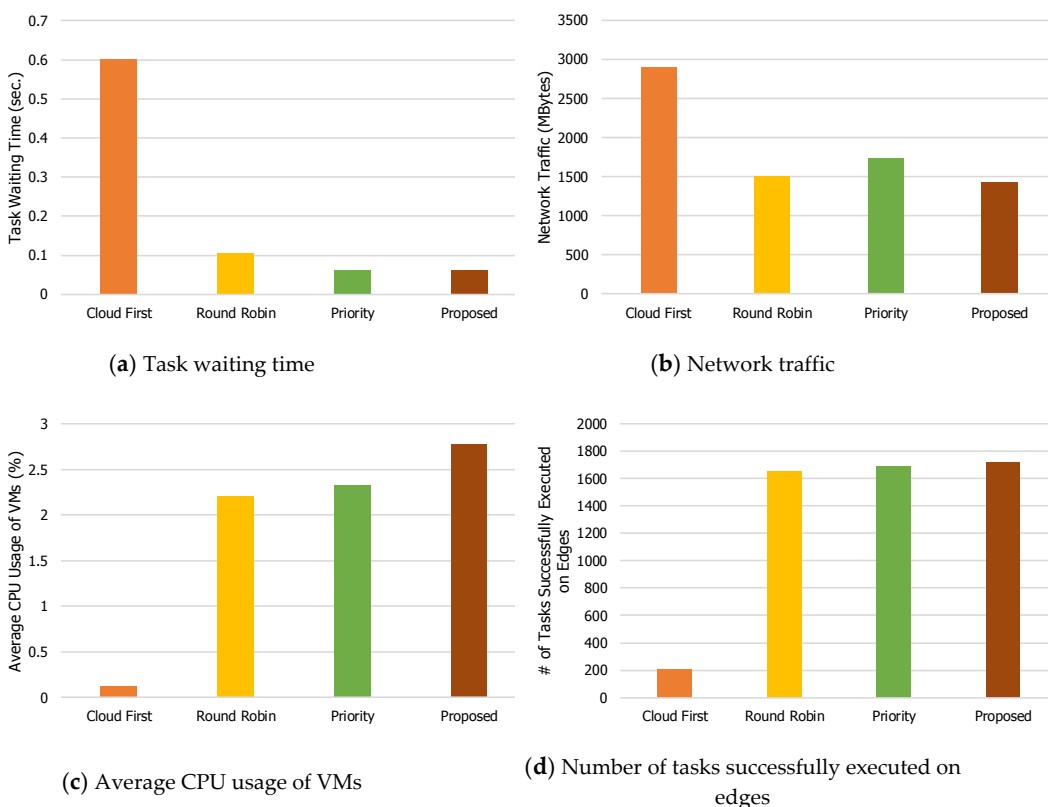

(**a**) Task waiting time

(**b**) Network traffic

(**c**) Average CPU usage of VMs

(**d**) Number of tasks successfully executed on edges

**Figure 5.** Performance comparisons when the number of mobile devices is 400.

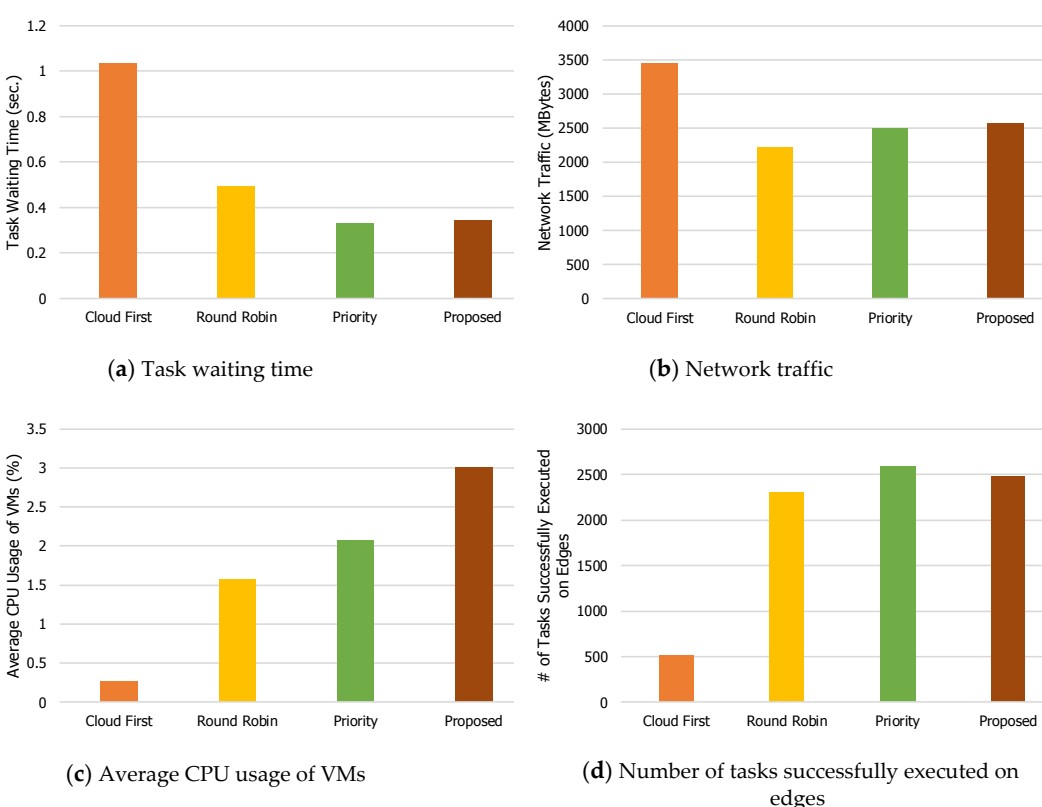

(**a**) Task waiting time

(**b**) Network traffic

(**c**) Average CPU usage of VMs

(**d**) Number of tasks successfully executed on edges

**Figure 6.** Performance comparisons when the number of mobile devices is 600.

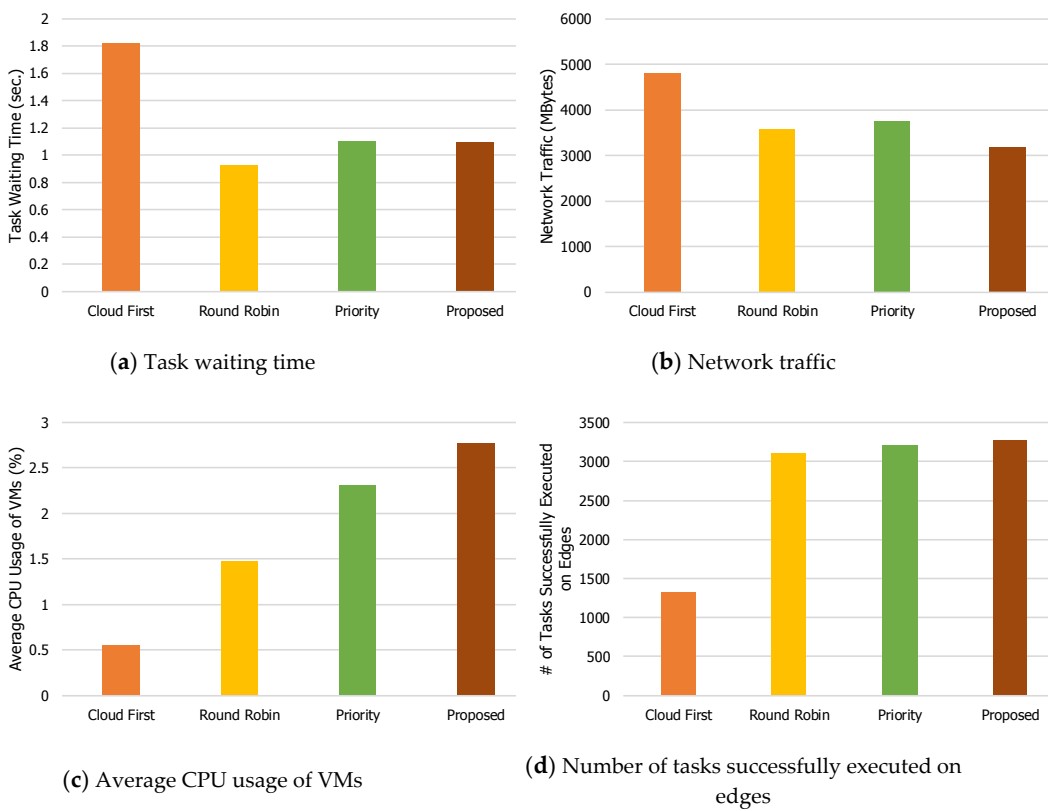

(**a**) Task waiting time

(**b**) Network traffic

(**c**) Average CPU usage of VMs

(**d**) Number of tasks successfully executed on edges

**Figure 7.** Performance comparisons when the number of mobile devices is 800.

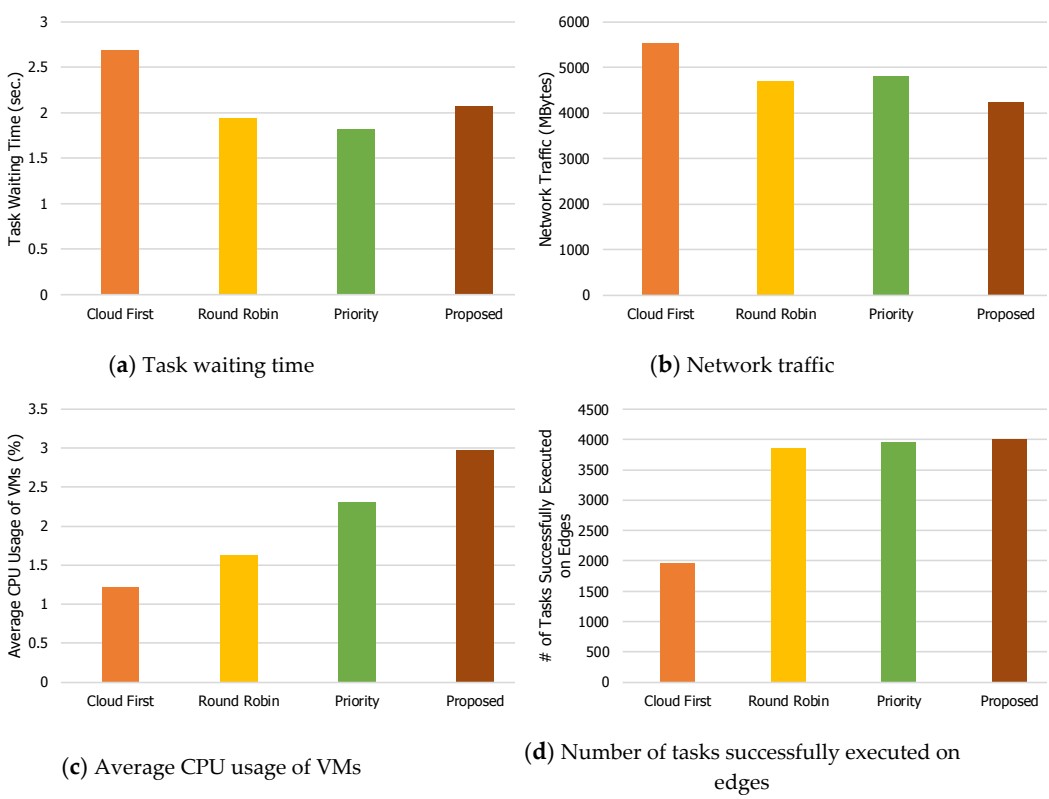

(**a**) Task waiting time

(**b**) Network traffic

(**c**) Average CPU usage of VMs

(**d**) Number of tasks successfully executed on edges

**Figure 8.** Performance comparisons when the number of mobile devices is 1000.

For network traffic (Figure 4b), the Cloud First method generates the largest traffic (1448 MB) because it mostly uses wide area networks, which are prone to failures and tend to try retransmission

for data and application files. The network traffic for the other three techniques is 666 MB, 816 MB, and 615 MB for Round Robin, Priority, and Proposed, respectively. Our proposed load balancing technique generates the least network traffic since it outdoes the others in terms of load balancing, and therefore, the probability of network failures is lower than with the other three techniques.

The average CPU usage of virtual machines on edge servers indicates how well it utilizes edge servers. Figure 4c shows that the average CPU usage of virtual machines on edge servers when the number of mobile devices is 200. Because the Cloud First method prefers to use the central cloud server, it shows that the average CPU usage of virtual machines on edge servers is very low (0.13%), that for the proposed load balancing technique is 2.9%, and those for Round Robin and Priority are 2.14% and 2.78%, respectively. This shows that the proposed load balancing technique utilizes edge servers better than the other three techniques.

Figure 4d shows the number of tasks successfully executed on edge servers. Not surprisingly, the proposed technique has the highest results and the Cloud First method has the lowest. Accordingly, the number of tasks successfully executed on the central server for the Cloud First method is the highest (not shown in the figure). That is, the Cloud First method does not utilize edge servers and generates more network traffic for offloading tasks.

The phenomenon shown in Figure 4 can also be seen in Figures 5–8, with some exceptions. For example, when the number of mobile devices is 600 (Figure 6), the network traffic of the Round Robin method is the lowest. The reason is that task failures for the Round Robin method are increased, mainly due to the lack of guarantees in the delay time. As for the number of tasks successfully executed on edge servers when the number of mobile devices is 600 (Figure 6d), the Priority method has the largest value.

Priority tries edge servers even when edge servers cannot execute the offloaded tasks. The average CPU usage of VMs (Figure 6c) supports this. The average CPU usage of VMs for the Priority method is 2.07%, while that for the proposed technique is 3.0%. This indicates that the proposed load balancing technique utilizes both edge servers and the central cloud server. In other words, when edge servers cannot support the offloaded tasks, the proposed technique forwards them to the central cloud server.

For the task waiting time (Figures 7a and 8a), the proposed load balancing technique has a higher value than other technique(s). The reason that the proposed technique has a slightly higher value is that the proposed technique checks the waiting time for edge servers as well as the forwarding time for offloading tasks from edge servers to the central cloud server, as described in Section 3. Again, when the waiting time for edge servers is less than the forwarding time for offloading tasks from edge servers to the central cloud server, the proposed technique does not forward the offloaded tasks to the central cloud. Otherwise, it forwards the offloaded tasks to the central cloud server. This helps the overall performance while reducing the waiting time for offloading tasks.

## 5. Conclusions

Providing scalability to data centers and servers is of great concern and a few load balancing techniques have been suggested by previous studies in edge cloud computing environments. To incorporate a load balancing technique with scalability into edge cloud computing environments, we proposed a load balancing algorithm that effectively distributes offloaded tasks from a hot spot to nearby edge servers based on a graph coloring theory with the implementation of a genetic algorithm that reduces the algorithm's complexity. Performance results show that the proposed load balancing algorithm outperforms previous techniques and increases the average CPU usage of virtual machines, which indicates a high utilization of edge servers. Compared with techniques in previous studies (Cloud First, Round Robin, and Priority), the proposed technique generates the least network traffic, and thus, the probability of network failures is lower than with the other three techniques. As far as the utility of edge servers and the central server is concerned, our method utilizes both edge servers and the central cloud server well, such that when edge servers cannot support the offloaded tasks, the proposed technique forwards them to the central cloud server. Future work will include adding

failure models for mobile devices and edge servers and predicting various types of failure based on artificial intelligence and machine learning techniques to further improve the load balancing metrics.

**Author Contributions:** Methodology and writing-original draft preparation, J.L.; conceptualization and writing-review, D.L. as the corresponding author. All authors read and approved the final manuscript.

**Funding:** This work was supported by the National Research Foundation of Korea Grant funded by the Korean Government (NRF-2019R1H1A1100790 and NRF-2018R1D1A1B07045838).

**Acknowledgments:** We thank the anonymous reviewers for their careful reading and insightful suggestions to help improve the manuscript.

**Conflicts of Interest:** The authors declare no conflict of interest.

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
