# Peer review of "A Load Balancing Algorithm for Mobile Devices in Edge Cloud Computing Environments"

_electronics, doi:10.3390/electronics9040686_

Round 1
Reviewer 1 Report
This paper describes a novel load balancing algorithm for edge cloud computing. In particular, the paper focuses on the problem of mobile devices offloading tasks to edge servers.
The paper is well written and clearly understandable.
In this reviewer’s opinion, only two points should be clarified before publication:
1) in the “Performance Evaluation” Section, authors describe the performed experiments and the measurements. In particular, on page 7, authors wrote that they increased the number of mobile devices from 200 to 1000 with an increment of 200 and each mobile device generated from 5 to 9 tasks. Why did the authors choose to generate from 5 to 9 tasks? How are the tasks generated? How are the tasks distributed? This reviewer suggests to insert more details about this important aspect.
2) in Section 2.2, authors briefly described genetic algorithms. In particular, Figure 3 has many blocks, but the description is too short. The suggestion is to better describe figure 3 in order to improve the quality of the manuscript and allow non-expert in genetic algorithms to clearly understand the content of the paper.
Minor points and typos:
- at line 38, please change “require” with “requires”
- please, rephrase lines 40-41
- at line 89, please insert “of” between “are” and “the same color”
- at lines 127-128 please change “mobile device discoverable” with “discoverable mobile device”
- please, rephrase lines 134-135
- at lines 197, please change “Figure” with “Figures”
Author Response
We appreciate the anonymous reviewers for their careful reading of our manuscript and their many insightful comments and suggestions. Below we respond to the comments of the reviewer in detail. We are also providing a revised manuscript that reflects their suggestions and comments. We feel that it has resulted in a stronger manuscript.
Reviewer 1's Comments:
1) in the “Performance Evaluation” Section, authors describe the performed experiments and the measurements. In particular, on page 7, authors wrote that they increased the number of mobile devices from 200 to 1000 with an increment of 200 and each mobile device generated from 5 to 9 tasks. Why did the authors choose to generate from 5 to 9 tasks? How are the tasks generated? How are the tasks distributed? This reviewer suggests to insert more details about this important aspect.
=> Thank you for insightful comments. To reflect, we added the following sentences to Section 4.
For types of tasks, we consider three applications, that is, augmented reality, health monitoring, and computation-intensive data processing. Since types of the three applications are different, properties of the three applications are also dissimilar. For example, the augmented reality application requires low latency, the health monitoring application sends sensor data and the server examines (processes) the data and forwards results to the application timely, and the computation-intensive data processing application sends programming codes to the server and the server returns results afterward. Regarding this, we configure a mobile device to generate random tasks from 5 to 9 at a time according to the types of applications [30]. All the tasks from mobile devices in a hotspot are distributed according to the policies of the algorithms; in our case, the tasks are distributed to nearby edge servers up to the number of colors evenly.
2) in Section 2.2, authors briefly described genetic algorithms. In particular, Figure 3 has many blocks, but the description is too short. The suggestion is to better describe figure 3 in order to improve the quality of the manuscript and allow non-expert in genetic algorithms to clearly understand the content of the paper.
Minor points and typos: …
=> Thank you for pointing out our missing contents. We added the following sentences to Section 2.2 and corrected typos you indicated.
Specifically, a genetic algorithm can be used to solve optimization problems for large datasets and search space. Unlike other strategies, a genetic algorithm uses a random approach starting from random initialization and it tries to find a solution by searching only in the space based on fitness scores. The fitness function is used for scoring each chromosome (solution candidate) and individuals that have best fitness scores will be used for reproduction.
To survive from the fitness, individuals try to find the region, in which chances of getting the better fitness score. Then, it generates an offspring (reproduction) by performing crossover and mutation processes. In the crossover process, a pair of parents generates an offspring by selecting random points from the parents, while in the mutation process, some of genes may have different values from the parents by mimicking the natural mutation. The reproduction process is iterated until the termination condition is met.

Reviewer 2 Report
The article deals with the issue of communication between mobile devices and edge servers within the cloud computing. It is a combinatorial task, the matter of which is to search a large state space. The authors solve this problem by graph theory by coloring the nodes of the equivalent graph. It is known that this task is an N-P completely problem for which there is no effective algorithm yet. The core of the work is the design of an heuristic algorithm based on the evolutionary principle (Chapter 3).
The methodology is described clearly, the experimental results are sufficiently presented.
I only have comments on the Conclusions chapter, which I think is too brief. It should include, for example, discussion and comparison with similar methods and highlight the advantages or disadvantages of the proposed solution.
Author Response
We appreciate the anonymous reviewers for their careful reading of our manuscript and their many insightful comments and suggestions. Below we respond to the comments of the reviewer in detail. We are also providing a revised manuscript that reflects their suggestions and comments. We feel that it has resulted in a stronger manuscript.
Reviewer 2's Comments:
I only have comments on the Conclusions chapter, which I think is too brief. It should include, for example, discussion and comparison with similar methods and highlight the advantages or disadvantages of the proposed solution.
=> Thank you for pointing out the weakness of the manuscript. To reflect your warm comment, we added the following sentences to the Conclusions section.
Compared with previous studies (Cloud First, Round Robin, and Priority), the proposed technique generates the smallest network traffic and, thus, the probability of network failures is lower than the other three techniques. As far as the utility of edge servers and the central server is concerned, our method utilizes both edge servers and the central cloud server well, such that when edge servers cannot support offloading tasks, the proposed technique forwards them to the central cloud server.

Reviewer 3 Report
Authors of the work “A Load Balancing Algorithm for Mobile Devices in Edge Cloud Computing Environments” present in this paper a novel load balancing technique features an efficient complexity by a graph coloring-based implementation based on a genetic algorithm, which aim is to distribute offloading tasks to nearby edge servers in an efficient way
Overall, it is a well-structured paper; the introduction section is wide and presents the purpose of the research in detail (Load Balacing in Mobile Devices in EC). There is a comparison and evaluations of the proposed method, using proper figures and showing up the proofs of the experiments included in the paper.
Although the proposal is interesting and within the scope of Electronics Journal, there are different issues that should be addressed in order to improve the work.
[Minor Comments]
- The state-of-the-art review could show the shortcomings of the works reviewed and how to address each of these shortcomings and how the proposed work addresses them.
- It is recommended to make a section with load balancing algorithms applied to mobile devices, in which their advantages and disadvantages are exposed. A comparison table should also be included.
- It is recommended that the article be reviewed by a native English speaker to improve some expressions.
Author Response
We appreciate the anonymous reviewers for their careful reading of our manuscript and their many insightful comments and suggestions. Below we respond to the comments of the reviewer in detail. We are also providing a revised manuscript that reflects their suggestions and comments. We feel that it has resulted in a stronger manuscript.
Reviewer 3's Comments:
The state-of-the-art review could show the shortcomings of the works reviewed and how to address each of these shortcomings and how the proposed work addresses them. It is recommended to make a section with load balancing algorithms applied to mobile devices, in which their advantages and disadvantages are exposed. A comparison table should also be included.
=> Thank you for insightful comments. For the research review, we added the subsection 2.3 to the manuscript as below.
2.3. Literature Review
In [30], the authors proposed a hierarchical edge cloud architecture that distributes peak loads to different tiers of cloud servers. To adaptively distributed mobile workloads to different tiers, they developed a workload placement algorithm that monitors edge servers' computational capacity. In [31], the authors proposed a cooperative approach for two data centers. When a data center cannot accept a user's request, it forwards the request to another data center. The authors designed a mathematical model for the cooperative approach with a Poisson flow and suggested the dynamic tuning of system parameters.
In [32], it uses the graph repartitioning theory to implement a load balancing algorithm supporting dynamicity. The authors constructed the system model of fog computing and developed the load balancing technique by considering vertices as represent nodes and edges as data dependencies between tasks. Although the concept is similar to our approach using the graph theories, the implementation and the aim of load balancing differ. While our method is to distribute offloading tasks to nearby edge servers in an efficient way, the authors' approach is to minimize the migration overhead.
The authors of [33] developed an authentication technique in edge cloud computing environments. For task allocation, it tries to find less overloaded edge servers by monitoring edge servers in the system and performs the breadth first search method to implement the load balancing technique. While the authors showed the effectiveness of the load balancing technique, the scalability and the complexity issues of the algorithm remain unresolved.
Table 1 shows the feature comparison with the aforementioned studies. Our technique considers both workloads and edge servers for load balancing with mobility support. More importantly, the proposed technique scales well in terms of the size of mobile devices and the algorithm's complexity. When nearby edge servers are busy, our method forwards the request to the central cloud server. To minimize response time and support real-time applications, we consider nearby edge servers from the users and implement the algorithm based on the location of the mobile users.
Table 1. Feature comparison with previous studies.
Feature |
[30] |
[31] |
[32] |
[33] |
Proposed |
Workload load balancing |
™ |
™ |
™ |
™ |
™ |
Edge server load balancing |
Í |
™ |
Í |
Í |
™ |
Support mobility |
Í |
Í |
™ |
Í |
™ |
Scalability |
r |
Í |
Í |
Í |
™ |
Support both edge servers and the central cloud |
™ |
Í |
Í |
Í |
™ |
Support Real-time |
Í |
™ |
Í |
Í |
™ |
Location-aware |
Í |
Í |
™ |
Í |
™ |
It is recommended that the article be reviewed by a native English speaker to improve some expressions.
=> Thank you for the recommendation. We have consulted an English editing service for the issue and the corrections are marked as red text in the manuscript.
